# Biochar Suppresses Bacterial Wilt of Tomato by Improving Soil Chemical Properties and Shifting Soil Microbial Community

**DOI:** 10.3390/microorganisms7120676

**Published:** 2019-12-10

**Authors:** Yang Gao, Yang Lu, Weipeng Lin, Jihui Tian, Kunzheng Cai

**Affiliations:** 1College of Natural Resources and Environment, South China Agricultural University, Guangzhou 510642, China; likoscau@gmail.com (Y.G.); superloo@163.com (Y.L.); linweipeng1986@163.com (W.L.); jhtian@scau.edu.cn (J.T.); 2Key Laboratory of Tropical Agricultural Environment in South China, Ministry of Agriculture, Guangzhou 510642, China; 3Tea Research Institute, Guangdong Academy of Agricultural Sciences, Guangzhou 510640, China

**Keywords:** biochar, tomato, soilborne disease, soil chemical properties, bacterial community

## Abstract

The role of biochar amendments in enhancing plant disease resistance has been well documented, but its mechanism is not yet fully understood. In the present study, 2% biochar made from wheat straw was added to the soil of tomato infected by *Ralstonia solanacearum* to explore the interrelation among biochar, tomato bacterial wilt resistance, soil chemical properties, and soil microbial community and to decipher the disease suppression mechanisms from a soil microbial perspective. Biochar application significantly reduced the disease severity of bacterial wilt, increased soil total organic carbon, total nitrogen, C:N ratio, organic matter, available P, available K, pH, and electrical conductivity. Biochar treatment also increased soil acid phosphatase activity under the non-*R.*-*solanacearum*-inoculated condition. High-throughput sequencing of 16S rRNA revealed substantial differences in rhizosphere bacterial community structures between biochar-amended and nonamended treatments. Biochar did not influence soil microbial richness and diversity but significantly increased the relative abundance of Bacteroidetes and Proteobacteria in soil at the phylum level under *R. solanacearum* inoculation. Furthermore, biochar amendment harbored a higher abundance of *Chitinophaga*, *Flavitalea*, *Adhaeribacter*, *Pontibacter*, *Pedobacter*, and *Ohtaekwangia* at the genus level of Bacteroides and *Pseudomonas* at the genus level of Proteobacteria under *R. solanacearum* inoculation. Our findings suggest that a biochar-shifted soil bacterial community structure can favorably contribute to the resistance of tomato plants against bacterial wilt.

## 1. Introduction

Bacterial wilt, caused by *Ralstonia solanacearum*, is a bacterial soilborne disease that is commonly observed among *Solanaceous* crops [1]. This disease is likely to occur under high temperatures and humid conditions and can persist for a long time [2]. The traditional methods for controlling bacterial wilt mainly focus on resistant species, biological control, chemical control, and soil anaerobic disinfection [3,4,5]. However, these methods may have limited or negative effects on food safety and the environment. Therefore, more effective and ecofriendly approaches need to be developed to control this disease.

Biochar is a refractory and highly aromatized carbonaceous solid that is produced by the high-temperature slow pyrolysis (usually <700 °C) of biological residues in the absence of oxygen [6]. Biochar has several major constituents, including hydrogen and oxygen, and is especially rich in carbon (about 70–80%) [7]. Many studies have shown that biochar can improve soil fertility [8,9,10], affect the physical and chemical properties of soil [11,12], and increase the soil adsorption of heavy metal ions [13,14]. Biochar also plays an important role in enhancing the disease resistance of plants [15,16]. Previous studies found that biochar amendments could promote the systemic resistance of strawberry plants against *Botrytis cinerea*, *Colletotrichum acutatum*, and *Podosphaera apahanis* [17] as well as reduce foliar disease in tomatoes and green peppers caused by *B. cinerea* and *Oidiopsis sicula* [18]. Applying biochar could also alleviate the plant diseases caused by soilborne pathogens [19,20,21] and significantly reduce the incidence of *Fusarium* wilt in tomato [22], as well as bacterial wilt in both tomato [23] and tobacco [24]. 

Evidence shows that biochar suppresses pathogens mainly by promoting the systemic resistance of plants, improving soil and plant nutrition, regulating soil microflora, and adsorbing and detoxifying chemicals [17,25,26,27,28,29]. Some reports revealed that biochar control soilborne diseases by stimulating the growth of beneficial microorganisms [30,31]. However, the effects of biochar on plant disease may vary with different biochar feedstocks, application dose, soil type, and disease types [15,20,26]. The potential mechanisms of biochar in enhancing plant resistance from a microbial perspective are not fully understood [15].

Crop residues are one of the major source materials for producing biochar by pyrolysis. Our previous study revealed that wheat and peanut biochar has a significant role in enhancing tomato resistance against bacterial wilt, and wheat biochar showed better effects [32]. In this study, we further investigated the effects of wheat biochar application on soil chemical properties and enzymatic activity. Illumina MiSeq was performed to identify the impacts of biochar on soil microbial communities under *R. solanacearum* infection and to decipher the possible mechanisms of biochar-mediated bacterial wilt resistance of tomato. 

## 2. Materials and Methods

### 2.1. Biochar and Soil Characteristics

Wheat biochar (produced by Shangqiu Sanli New Energy Co., Ltd., Shangqiu, China) was used in the present study. The characteristics of wheat biochar are as follows: pH 10.0, total C 47%, total N 1%, ash content 48%, available P 325 mg·kg^−1^, available K 3820 mg·kg^−1^, and C:N ratio 47. 

The soil used in this study was sandy loam (Eutric Cambisol) collected from a field in Zhucun Village, Zengcheng City, Guangdong, China. The chemical properties of the soil are as follows: 16.3 g·kg^−1^ soil organic matter (SOM), 116 mg·kg^−1^ available N, 151 mg·kg^−1^ available P, 83 mg·kg^−1^ available K, 0.9 g·kg^−1^ total N, 1.3 g·kg^−1^ total P, 29 g·kg^−1^ total K, pH 5.9, and 9.8 C:N ratio.

### 2.2. Plant Growth and R. Solanacearum Culture

Tomato plants (cv. Taiwan red cherry, produced by Kefeng Seed Co., Ltd., Changchun, China) susceptible to *R. solanacearum* were used in the experiment. Tomato seeds were stored in a refrigerator at 4 °C and immersed in water at room temperature for 2 h before use. Afterwards, the seeds were surface sterilized in water at 50 °C for 15 min and were placed on moist filter paper in Petri dishes. After 2 days, the germinated seeds were transferred in nursery soil (disinfected at 150 °C for 4 h) in a growth chamber with 14 h light exposure (light intensity 200 μmol·m^−2^·s^−1^) at 30 °C/25 °C (day/night). After 2 weeks, the tomato plants were transplanted to polyethylene plastic pots (170 × 165 mm) filled with 2 kg of soil. Each pot had two tomato plants that were maintained at 28 °C in a greenhouse for 3 weeks until the end of the experiment. 

A moderate pathogenic *R. solanacearum* strain *RsH* (race biovar 3, provided by the College of Horticulture, South China Agricultural University, Guangzhou 510642, China) was used in the experiment. *R. solanacearum* was cultured in LB media at 30 °C for 48 h, harvested from agar plates via sterile water flushing, and adjusted to OD_600_ = 0.06 (approximately 10^8^ CFU mL^−1^) as a bacterial inoculum suspension [33]. At the sixth leaf stage of the tomato plants, 50 mL of bacterial inoculum suspension was poured into each pot over the soil surface.

### 2.3. Experiment Design

There were four treatments in the experiment, including CK (no biochar and no *R. solanacearum* inoculation), Rs (*R. solanacearum* inoculation without biochar amendment), BC (2% wheat straw biochar addition without *R. solanacearum* inoculation), and Rs + BC (2% wheat straw biochar addition and *R. solanacearum* inoculation). The experiment was arranged in a completely randomized design with three replications. The experiment was conducted twice with 10 plants for each biological replicate for disease severity analysis (representative data from a single experiment for analysis). When the sixth leaf of the tomato plant appeared, the roots of each tomato plant were lightly stabbed and inoculated with *R. solanacearum* by pouring 10 mL of the bacterial suspension into each pot. The roots of the non-inoculated tomato plants in CK were also stabbed with the same volume of deionized water as control.

### 2.4. Disease Severity Survey

Before collecting soil samples, disease severity was determined for these two pathogen-inoculated treatments (Rs, Rs + BC) based on the methods described by Fang [34]: (0) no or slight illness of the plant; (1) wilting of less than 25% of the plant; (2) wilting of about 26–50% of the plant; (3) wilting of about 51–75% of the plant (systemic wilting); (4) wilting of about 75–100% of the plant (systemic wilting); (5) plant death.

### 2.5. Soil Sampling and Analysis

Soil samples of different treatments were collected from all pots in a replicate at the end of the experiment (7 days after the *R. solanacearum* inoculation). Bulk soil was collected from each pot in the same treatment and composited into one bag, then divided into two parts; one was air-dried for chemical analysis, and another was stored at 4 °C for assessment of soil enzymes. The rhizosphere soil samples of different treatments were collected and transported in ice chests to the laboratory immediately for DNA extraction for soil microbial community analysis.

Soil pH was measured by adding 25 mL of deionized water to 10 g of soil using a pH meter (PHB-3, SANXIN, Shanghai, China). EC (µS·cm^−1^) was determined with a microprocessor conductivity meter (DDS-307, YUEPING, Shanghai, China) by taking soil and distilled water (*w*/*v*) in a 1:5 ratio. Total organic carbon (TOC), total nitrogen (TN), and C:N ratio (C/N) were analyzed using an automated TOC analyzer (Vario TOC, Elementar, Langenselbold, Germany). Soil organic matter was analyzed by using a K_2_Cr_2_O_7_-H_2_SO_4_ solution with FeSO_4_ titration [35]. Alkali-hydrolyzable N was determined with 1.0 M NaOH extraction according to Lu [36] as the indicator of N availability. Available P was extracted by 0.5 M NaHCO_3_ solution following the method of Olsen [37]. Available K was extracted by 1 M NH_4_OAc (pH 7) solution and analyzed using an atomic absorption spectrophotometer (novAA350, Analytik Jena, Jena, Germany).

The method described by Zantua and Bremner [38] was used for urease activity measurement with urea solution as a substrate; determination of the ammonium released by urease activity when soil is incubated with tris (hydroxymethyl) aminomethane (THAM) buffer, urea solution, and toluene at 37 °C for 2 h; and assayed colorimetrically at 578 nm (TU-1901, PERSEE, Beijing, China). Acid phosphatase was determined according to the method described by Tabatabai and Bremner [39], which involves colorimetric estimation at 400 nm of the p-nitrophenol released by phosphatase activity when soil is incubated with buffered (pH 6.5) sodium p-nitrophenyl phosphate solution and toluene at 37 °C for 1 h (TU-1901, PERSEE, Beijing, China). Catalase activity was assessed by monitoring the decrease caused by H_2_O_2_ consumption and using its molar extinction coefficient (ε = 36 M^−1^ cm^−1^) in the absorbance at 240 nm (TU-1901, PERSEE, Beijing, China) following the Havir and McHale [40] method.

### 2.6. Soil DNA Extraction, PCR Amplification, and Illumina MiSeq Sequencing

Soil DNA was extracted from 0.5 g of fresh soil by using the FastDNA^®^ Spin Kit for Soil (MP Biomedicals, Irvine, CA, USA) according to the manufacturer’s instructions. The extracted products were then stored at −20 °C. To exhibit few biases against different bacterial taxa, a set of bacterial primers—515F (5′-GTGCCAGCMGCCGCGGTAA-3′) and 806R (5′-GGACTACHVGGGTWTCTAAT-3′) [41]—were selected to amplify the hypervariable V4 regions of 16S rRNA. The length of the amplified fragment was approximately 256 bp. PCR reactions were performed in triplicate in a final volume of 30 μL mixture containing 0.75 units of Ex Taq DNA polymerase (TaKaRa, Dalian, China), 1 × Ex Taq loading buffer (TaKaRa, Dalian, China), 0.2 mM dNTP mix (TaKaRa, Dalian, China), 0.2 µM of each primer, and 100 ng template DNA. The PCR products were extracted from a 2% agarose gel, mixed in equimolar, and further purified by using a gel extraction kit. The PCR conditions were as follows: 3 min at 95 °C, followed by 35 cycles of 94 °C for 30 s, 1 min at 50 °C, 1 min at 72 °C, and 10 min at 72 °C.

The PCR products of the soil samples were sequenced based on Illumina MiSeq paired end (PE) 300 bp. The following data denoising procedure was performed: the quality of the original data was controlled by using a homemade Perl script for the denoising procedure. The sequence reads had an average quality score less than 20 and read length less than 200 bp, and ambiguous bases and long stretches of homopolymers (more than 7 bp) were removed from the dataset. After reversing the barcode sequence, the double-stranded sequence was spliced by using Mothur to obtain the complete 16S rRNA gene V4 region sequence. Based on the barcode sequence for each sample on its PCR primers (806R), sequences were clustered and assigned to operational taxonomic units (OTUs) using the QIIME implementation of cd-hit with a threshold of 97% pairwise identity [42]. Representative OTUs were subsequently classified using the Green Gene database [43]. The rarefaction curves were created based on the Shannon diversity index and Sobs to compare the relative levels of OTU richness across all soil samples. For alpha diversity, the Chao1 index and ACE were calculated to characterize the richness (i.e., the number) of phenotypes of microbial communities in the soil samples [44]. Shannon and Simpson indices were used to estimate the diversity of the microbial communities in each soil sample [45]. As shown in the OTU table, we used the Jest method to calculate the matrix of the two distances of the microbial community structure of the sample, and, based on this matrix, we performed a Principal Coordinates Analysis (PCoA) to explore the differences in these community structures across different treatments. The samples were analyzed by using the unweighted group mean method. Based on the abundance of species, the number of minimum sample reads was calculated by using the Jest method to calculate the distance matrix of samples, and sample clustering was performed by using the approximately maximum likelihood phylogenetic trees constructed by FastTree.

### 2.7. Statistical Analysis

All data were plotted and calculated using Excel 2013 and entered into the SPSS 17.0 (SPSS, Chicago, IL, USA) statistical software for the one-way ANOVA. Significant differences among treatments (*p* < 0.05) were determined by Tukey’s multiple range test and Student’s *t*-test. The sample distance matrix was calculated by the Jest method, and PCoAs were performed using Unifrac. Chao1, ACE, Simpson and Shannon’s index was performed with Mothur (version 1.22.2). The correlation of soil chemical properties and soil bacterial community was based on the Pearson correlation coefficients. Redundancy analysis was performed, and a Monte Carlo permutation test (499 permutations) was used to test the significance of the first and second axes. RDA was performed using CANOCO 5.0 software.

## 3. Results

### 3.1. Disease Severity of Bacterial Wilt

Wheat biochar application significantly reduced bacterial wilt severity and increased tomato resistance (Figure 1). Compared with non-biochar treatment (Rs), biochar treatment (Rs + BC) delayed disease development by 2 days, the level of disease severity was less than 2 (wilting of about 26–50% of the plant) through the whole pathogen infection process, and no dead plant was observed in biochar-treated plants. However, most plants wilted entirely at 7 days post inoculation in non-biochar treatment.

### 3.2. Effects of Biochar on Soil Chemical Properties

Biochar amendment significantly increased soil TOC, TN, C/N, SOM, pH, and electrical conductivity (EC), regardless of *R. solanacearum* (Table 1). TOC, TN, C/N, SOM, pH, and EC in biochar-treated soil were changed by 1.53-, 1.26-, 1.51-, 1.53-, 1.10-, and 3.45-fold, respectively, for non-*R.*-*solanacearum* inoculation (BC) compared with CK, and 1.44-, 1.13-, 1.45-, 1.46-, 1.10-, and 3.29-fold, respectively, for *R. solanacearum* inoculation (Rs + BC) compared with Rs.

Biochar amendment significantly increased soil available P and K regardless of *R. solanacearum* and increased available N under *R. solanacearum* inoculation (Table 2). Especially, biochar increased available P and K by 1.07- and 4.04-fold for non-*R.*-*solanacearum* inoculation (BC), respectively, compared with CK. Biochar also increased available soil N, P, and K by 1.41-, 1.06-, and 4.99-fold for *R. solanacearum* inoculation (Rs + BC), respectively, compared with Rs.

### 3.3. Effects of Biochar on Soil Enzyme Activities

Generally, biochar application increased the activity of soil urease, acid phosphatase, and catalase (Figure 2). Urease and acid phosphatase activities in biochar-treated soil were increased by 1.19- and 1.18-fold, respectively, for non-*R.*-*solanacearum* inoculation (BC) compared with CK. Urease, acid phosphatase, and catalase activities were increased by 1.12-, 1.13-, and 1.24-fold, respectively, for *R. solanacearum* inoculation (Rs + BC) compared with Rs.

### 3.4. Effects of Biochar Application on Soil Bacterial Taxonomic Richness

A total of 1,187,827 pairs of original sequences were obtained from the Illumina MiSeq PE 300 bp sequencing of the PCR products of soil samples. Quality control was performed, and the 3′ end of the reads showed a significant decrease in the base quality. After removing the singleton OTUs, 12,588 OTUs were obtained from the 12 soil samples (three replications for each treatment).

We randomly took 20,000 effective sequences in each soil sample and calculated the alpha diversity data of each sample (Appendix A). The Chao index and ACE were used to estimate richness, while the Shannon and Sampson indices were calculated to evaluate functional diversity (Table 3). The results showed that biochar application did not influence the soil microbial diversity index, including the Shannon index, Simpson index, ACE, and Chao, regardless of *R. solanacearum* inoculation.

### 3.5. Effects of Biochar Application on Soil Bacterial Community Composition

Based on the basal quality control, a total of 12,588 bacterial sequence reads were obtained from 12 samples. The soil samples from different treatments showed that among the 12,588 OTUs, 8114 (64%) were accurate at the phylum level, while 7009 OUTs (55%) were accurate at the class level. Figure 3 presents the phylogenetic distances between the groups in the different treatments, showing the difference between bacterial communities. The samples from CK and Rs treatments were clustered together, which were far away from BC and Rs + BC treatments. The group with biochar treatment only was far away from the group with both biochar and *R. solanacearum* treatment, indicating that the samples had big differences in bacterial community structure. These findings suggest that the biochar amendment led to a shift in the bacterial community structure.

Detailed phylogenetic analyses grouped the rhizosphere-associated bacterial sequences across different treatments into 25 phyla (Appendix A). The 11 most abundant phyla types are shown in Figure 4. Proteobacteria, Firmicutes, Acidobacteria, Actinobacteria, Bacteroidetes, Planctomycetes, Verrucomicrobia, Gemmatimonadetes, Chloroflexi, and Nitrospira were the dominant bacterial phyla, accounting for more than 80% of the bacterial sequences. The relative abundances of Acidobacteria, Bacteroidetes, Proteobacteria, Nitrospira, and Gemmatimonadetes at the phyla level were significantly altered under different treatments (Figure 5). Biochar treatment (Rs + BC) significantly increased the relative abundance of Bacteroidetes compared with no-biochar treatment (Rs) under *R. solanacearum*, but relative abundances of Acidobacteria indicated the opposite. Biochar treatment (Rs + BC) improved the relative abundances of Proteobacteria, Nitrospira, and Gemmatimonadetes compared with treatment Rs, and the differences were significant compared with CK.

Bacteroidetes was especially enriched in soil with biochar application under *R. solanacearum* inoculation and further analyses were carried out at the genus level (Figure 6). The biochar treatment (Rs + BC) significantly influenced the dominant genus groups and significantly increased the relative abundance of *Chitinophaga*, *Flavitalea*, *Adhaeribacter*, *Pontibacter*, *Pedobacter*, *Ohtaekwangia*, and an unclassified genus compared with no biochar treatment (Rs) under *R. solanacearum* inoculation. Biochar amendment only (BC) significantly increased the relative abundance of *Parasegetibacter*, *Pontibacter*, and *Flavisolibacter* compared with CK and Rs treatments.

Proteobacteria was the most abundant phylum in all treatments (Figure 4) and further analyses were carried out at the genus level (Appendix A). Under *R. solanacearum*, biochar application significantly increased the relative abundance of *Pseudomonas* (Figure 7).

### 3.6. The Relationship between Soil Properties and Rhizosphere Bacterial Community

Redundancy analysis revealed that variation in bacterial community composition was significantly correlated with specific soil properties (Figure 8 and Appendix A). Bacteroidetes, Proteobacteria, Nitrospira, and Gemmatimonadetes were positively correlated with soil urease and acid phosphatase activities, pH, and available N and P. Acidobacteria presented contrasting behavior that was negatively correlated with soil urease and acid phosphatase activities, pH, and available N and P. The redundancy analysis and the further permutations test revealed that urease activity and pH are the key predictors of bacterial community structure. Soil available N, acid phosphatase activity, and available P also had a good relationship with bacterial community structure. Soil urease activity was the factor that had greatest influence on Proteobacteria, while soil available N had a dominant impact on Bacteroidetes.

## 4. Discussion

Biochar can enhance the resistance of plants against different diseases [17,18,46]. In the present study, wheat biochar amendment significantly reduced the disease severity of bacterial wilt caused by *R. solanacearum*, thereby confirming our previous findings that biochar can be used as an amendment to control bacterial wilt [32]. The mechanisms of biochar-mediated plant resistance against plant diseases may stem from several mechanisms, such as detoxifying the chemical agents, stimulating the nutrient uptake of plants, improving soil properties, influencing soil microbial communities, and inducing plant resistance [15,24,26]. Several studies have also reported that biochar can improve the physical and chemical properties of soil and, consequently, promote soil health [26,47,48,49]. Specifically, biochar addition increased the porosity, pH, cation exchange capacity, organic carbon, and available nutrients of soil but reduced its bulk density [24,50,51]. Our findings revealed that wheat biochar amendment resulted in higher levels of soil TOC, TN, C/N, SOM, pH, EC, and available P and K, regardless of pathogen inoculation (Table 1; Table 2), which is similar to the results of Wang et al. [52], who found that rice husk biochar promoted plant growth by improving the soil environment. Changes in soil pH from biochar amendment might be due to the strong basicity of biochar, and our results showed that soil pH has a positive relationship with total organic C; total N; and available N, P, and K (Appendix A). Moreover, the increase of available N, P, and K may influence the soil microbes and enzymatic activity, thus playing a vital role in the improvement of soil fertility [53].

Soil enzymes play an important role in the maintenance of soil health and nutrient cycling, which can be biological indicators of soil quality [52,53,54]. Wang et al. [52] demonstrated that soil invertase, urease, proteinase, neutral phosphatase, catalase, and polyphenol oxidase activities were enhanced by biochar application in apple replant soil. Another study by Ghani et al. [54] showed similar results. Our results showed that soil urease and acid phosphatase activities were enhanced with biochar amendment (Figure 2), and these two enzymes had a positive relationship with soil chemical properties such as available P and K (Appendix A). The increase of soil enzyme activity may enhance the viability of soil microbes and can accelerate nutrient and C cycling in the soil that is beneficial to plant health.

The composition and diversity of soil microbial communities are associated with soil quality and plant health [55,56]. Biochar application may also change the soil community structure and subsequently result in the antagonism, competition, or parasitism of microorganisms and pathogens in the soil to reduce the amount of pathogenic bacteria [26]. Our previous study explored that wheat and groundnut biochar-mediated bacterial wilt resistance were associated with the increase in soil bacteria and actinomycete densities and the reduction in soil fungi/bacteria and fungi/actinomycetes ratios [32]. Alpha diversity analysis of microbial communities revealed no significant differences in bacteria richness and diversity among different treatments (Table 3), which is consistent with the findings of Rutigliano et al. [57] and Luo et al. [58]. However, other studies showed that biochar application increases the overall diversity of soil bacteria communities [24,30,59,60]. Khodadad et al. [61] found that those soils treated with oak- and grass-derived biochar demonstrated losses in their microbial diversity. Several biochar properties, including pyrolysis temperature, adsorption ability, nutrient content, pH, water holding capacity, and hormone-like compounds, may also be related to the influence of this variation [15,20,27,28]. These above contradictory results indicate the complexity of the soil microbial process influenced by soil amendments.

Our analysis of microbial beta diversity revealed a different pattern among treatments (Figure 3). Biochar application significantly influenced the composition of the soil bacterial community (Figure 4) and reduced the relative abundance of Acidobacteria. The relative abundance of Acidobacteria was largely driven by soil pH at a continental scale, and low pH values could be more suitable for Acidobacteria [62]. Consistent with those of Kolton et al. [27] and Xu et al. [63], our findings revealed that the relative abundance of Bacteroidetes in biochar-treated soil under *R. solanacearum* infection (Rs + BC) significantly increased compared with the no biochar treatments (Figure 5). Bacteroidetes are generally specialized in the degradation of organic matter, promote plant growth, and increase their resistance to environmental stress [64,65]. Our results also indicated that the relative abundances of *Chitinophaga*, *Flavitalea*, *Adhaeribacter*, *Pontibacter*, *Pedobacter*, and *Ohtaekwangia* at the genus level, which are bioremediators of hydrocarbons [49], were significantly increased by biochar amendment (Figure 6). The changes in these genera may be related to plant resistance, but further studies must be conducted to explore the relationship between these genera and *R*. *solanacearum*. In the present study, biochar amendment significantly increased the relative abundance of *Pseudomonas* under the condition of *R. solanacearum* inoculation (Figure 7). *Pseudomonas* spp. are known for their ability to improve plant growth, suppress pathogens, and induce the systemic resistance of many plant species against diseases and pests [30,66,67]. Biochar has also been previously described as an agent for promoting plant growth or biocontrol. Mendes et al. [68] found that *Pseudomonas* microorganisms can resist the diseases caused by *R. solanacearum*, whereas other studies have used *Pseudomonas* spp. to biologically control *R. solanacearum* [69,70]. In sum, the addition of biochar may provide a suitable environment for promoting the growth of *Pseudomonas*, increasing the relative abundance of Pseudomonadaceae (thereby inhibiting the growth of *R. solanacearum*), and reducing the incidence of tomato bacterial wilt. A strong link was also observed among the biochar-induced changes in soil microbial environments, the suppression of soilborne diseases, and plant performance [30].

In conclusion, wheat biochar amendment significantly reduced the disease severity of bacterial wilt and increased tomato resistance. The addition of biochar increased soil pH and EC and improved soil nutrient status and enzymatic activity, while soil fertility and soil enzyme activity had a positive relationship with the abundance of the soil microbial community (Figure 8). Biochar addition did not influence soil bacterial microbial community diversity but significantly influenced the relative abundance of the soil microbial community at the phylum and genus levels, suggesting that biochar shifting the soil bacterial community structure contributed to pathogen resistance.

## Figures and Tables

**Figure 1 microorganisms-07-00676-f001:**
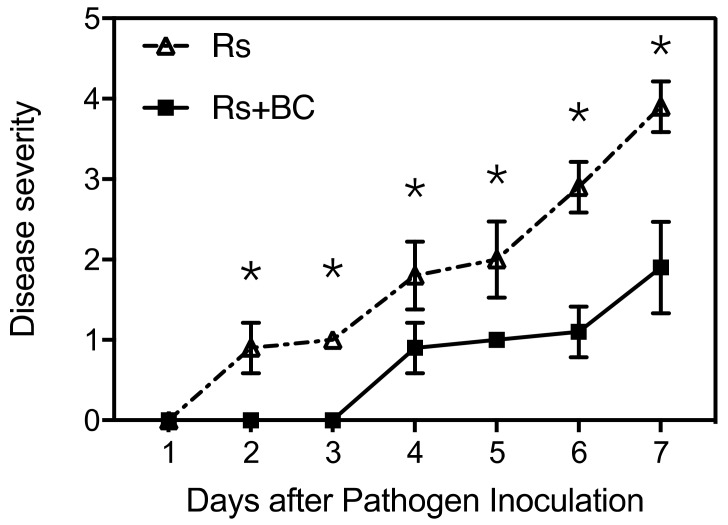
Effects of biochar on the disease severity of tomato bacterial wilt. Rs, *Ralstonia solanacearum* inoculation without biochar amendment; Rs + BC, biochar amendment and *R. solanacearum* inoculation. Disease severity was rated daily for 7 days using a disease severity scale, in which 0: no wilted leaves; 1: 1–25% leaves wilted; 2: 26–50% leaves wilted; 3: 51–75% leaves wilted, 4: >75% leaves wilted; and 5: plant death. The experiment was repeated twice with 10 plants for each biological replicate, * indicate significant difference between the two treatments (* *p* < 0.05, Student’s *t*-test).

**Figure 2 microorganisms-07-00676-f002:**
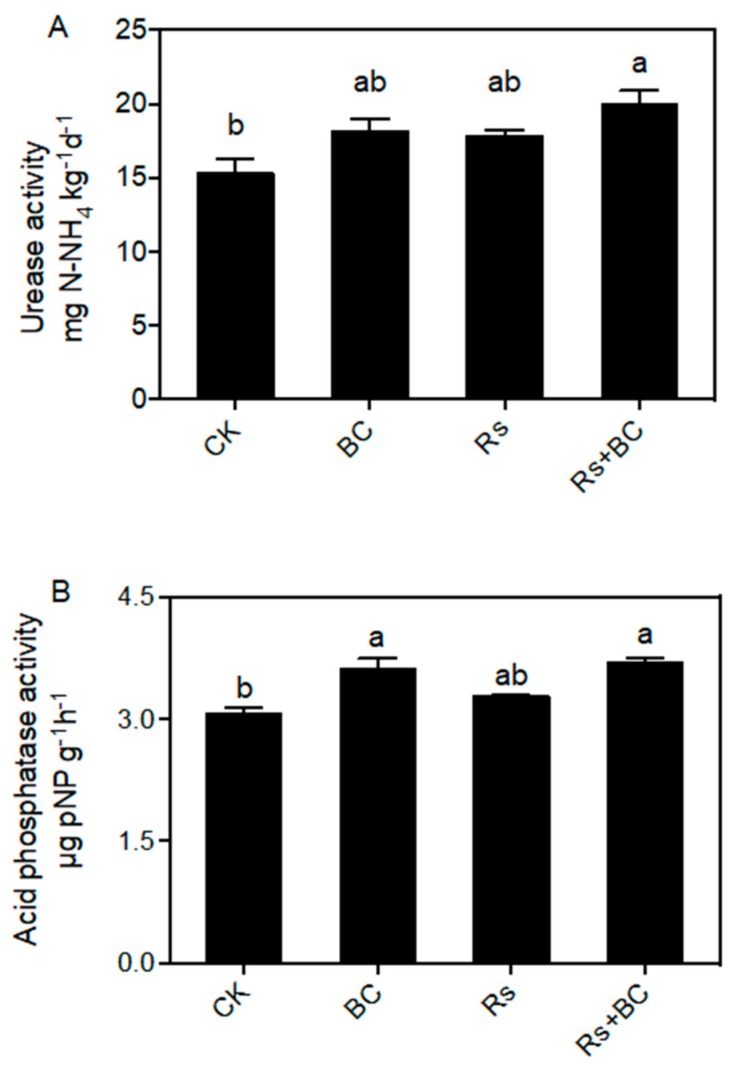
Effects of biochar and *R. solanacearum* inoculation on soil enzyme activities of Urease (**A**), Acid phosphatase (**B**) and Catalase (**C**). CK, no biochar and no *R. solanacearum* inoculation; Rs, *R. solanacearum* inoculation without biochar amendment; BC, biochar addition without *R. solanacearum* inoculation; Rs + BC, biochar amendment and *R. solanacearum* inoculation. Bars represent standard error. Columns with different letters indicate significant differences among treatments (*p* < 0.05).

**Figure 3 microorganisms-07-00676-f003:**
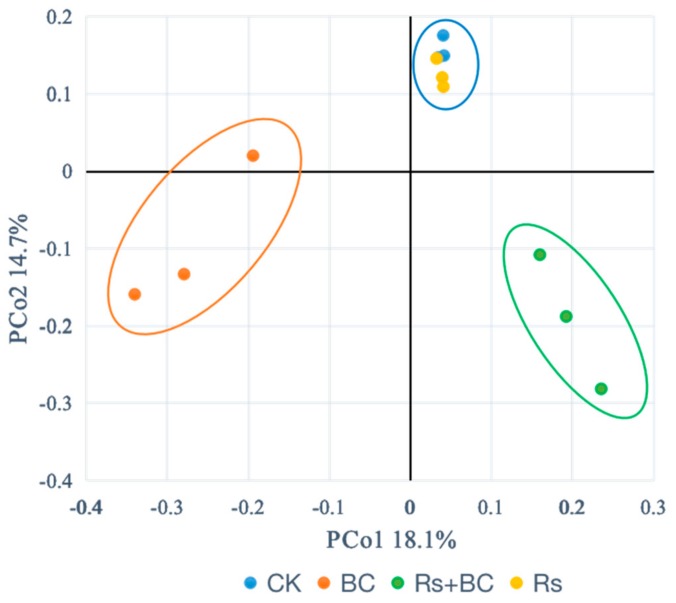
Phylogenetic distances of rhizosphere soil samples determined via Principal Coordinates Analysis (PCoA). CK, no biochar and no *R. solanacearum* inoculation; Rs, *R. solanacearum* inoculation without biochar amendment; BC, biochar addition without *R. solanacearum* inoculation; Rs + BC, biochar amendment and *R. solanacearum* inoculation.

**Figure 4 microorganisms-07-00676-f004:**
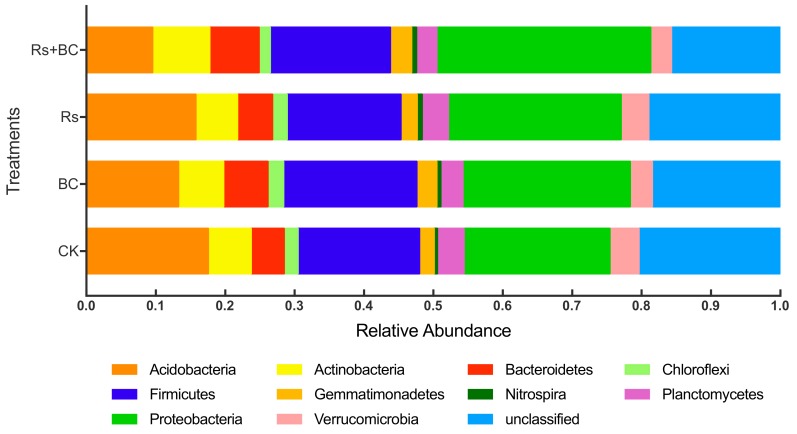
Effects of biochar and *R. solanacearum* inoculation on taxonomic distribution of operational taxonomic units (OTUs) (97% identity) at the bacteria phylum level of the rhizosphere soil (top 11). CK, no biochar and no *R. solanacearum* inoculation; Rs, *R. solanacearum* inoculation without biochar amendment; BC, biochar addition without *R. solanacearum* inoculation; Rs + BC, biochar amendment and *R. solanacearum* inoculation. The total sum of 16S rRNA gene sequence counts of OTUs among subsamples within plots and among replicate plots within each treatment was used to generate the relative abundance data for the figure.

**Figure 5 microorganisms-07-00676-f005:**
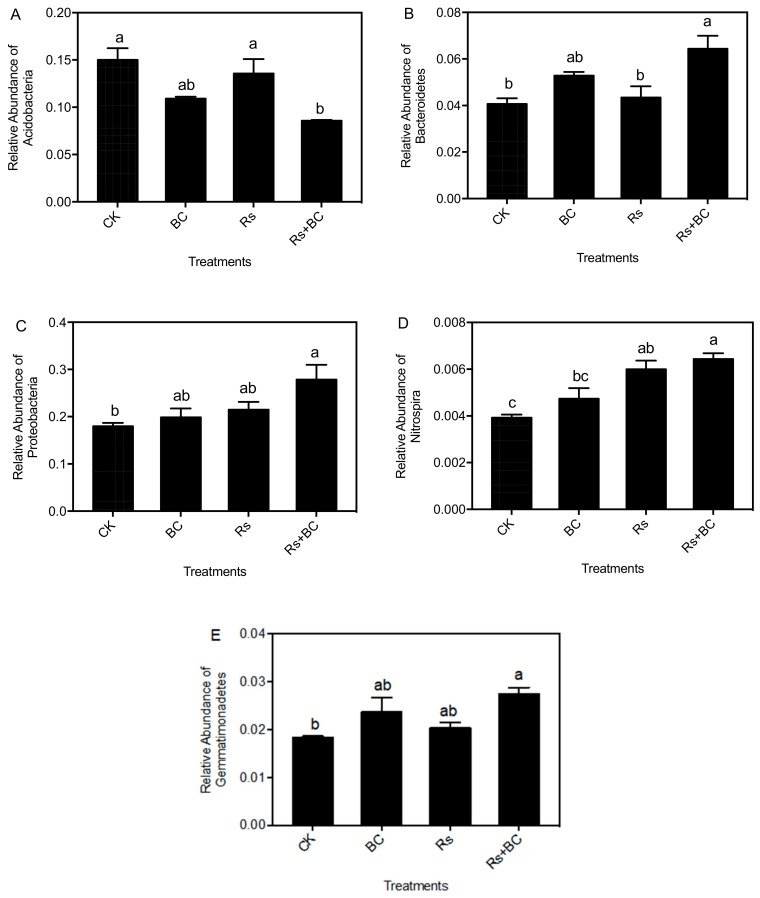
Relative abundance of Acidobacteria (**A**), Bacteroidetes (**B**), Proteobacteria (**C**), Nitrospira (**D**), and Gemmatimonadetes (**E**) of the rhizosphere soil. CK, no biochar and no *R. solanacearum* inoculation; Rs, *R. solanacearum* inoculation without biochar amendment; BC, biochar addition without *R. solanacearum* inoculation; Rs + BC, biochar amendment and *R. solanacearum* inoculation. Bars above the histogram represent the standard error of three replicates. Columns with different letters indicate significant differences among treatments (*p* < 0.05).

**Figure 6 microorganisms-07-00676-f006:**
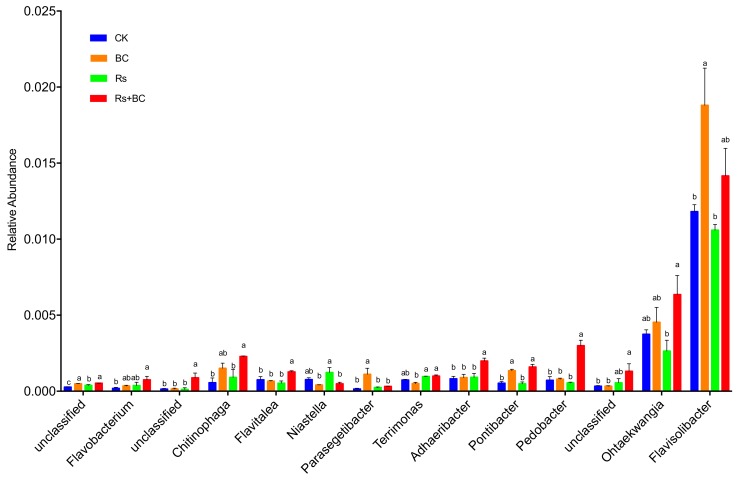
Relative abundance of Bacteroidetes in the rhizosphere soil amended with biochar and *R. solanacearum* inoculation. CK, no biochar and no *R. solanacearum* inoculation; Rs, *R. solanacearum* inoculation without biochar amendment; BC, biochar addition without *R. solanacearum* inoculation; Rs + BC, biochar amendment and *R. solanacearum* inoculation. Columns with different letters indicate different among treatments (*p* < 0.05).

**Figure 7 microorganisms-07-00676-f007:**
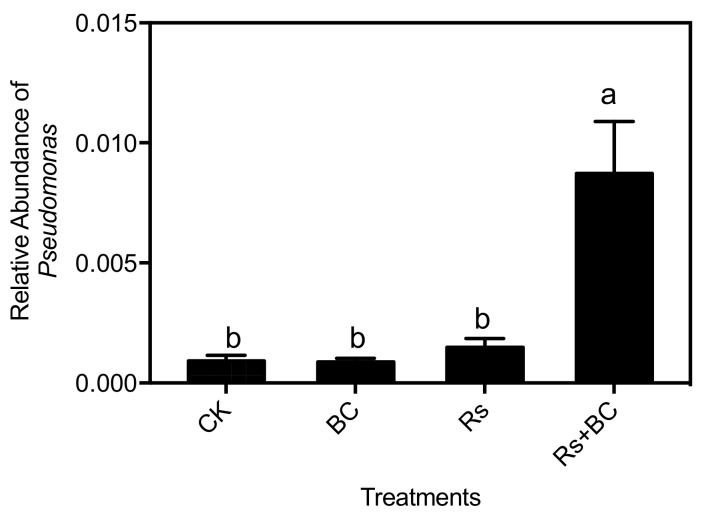
Relative abundance of *Pseudomonas* in rhizosphere soil amended with biochar and *R. solanacearum* inoculation. CK, no biochar and no *R. solanacearum* inoculation; Rs, *R. solanacearum* inoculation without biochar amendment; BC, biochar addition without *R. solanacearum* inoculation; Rs + BC, biochar amendment and *R. solanacearum* inoculation. Bars above the histogram represent the standard error of three replicates. Columns with different letters indicate significant differences among treatments (*p* < 0.05).

**Figure 8 microorganisms-07-00676-f008:**
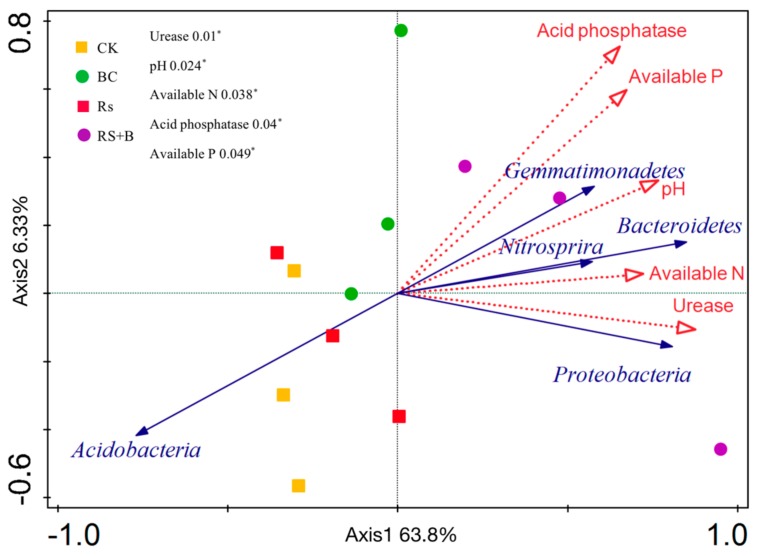
Redundancy analysis between soil properties and soil microbial community with biochar and *R. solanacearum* inoculation. CK, no biochar and no *R. solanacearum* inoculation; Rs, *R. solanacearum* inoculation without biochar amendment; BC, biochar addition without *R. solanacearum* inoculation; Rs + BC, biochar amendment and *R. solanacearum* inoculation. The soil properties were fitted to the plots using a permutations test (*p*-values).

**Table 1 microorganisms-07-00676-t001:** Effects of biochar and *R. solanacearum* on soil total organic carbon (TOC), total nitrogen (TN), C:N ratio (C/N), soil organic matter (SOM), pH, and electrical conductivity (EC).

Treatment	TOC g·kg^−1^	TN g·kg^−1^	C/N	SOM g·kg^−1^	pH	EC µS·cm^−1^
CK	7.3 ± 0.05 ^b^	0.85 ± 0.025 ^b^	7.16 ± 0.34 ^b^	12.4 ± 0.08 ^b^	5.9 ± 0.20 ^b^	111.3 ± 19.7 ^b^
BC	11.2 ± 0.56 ^a^	1.07 ± 0.005 ^a^	10.80 ± 0.4 ^a^	19.0 ± 0.95 ^a^	6.5 ± 0.05 ^a,b^	384.0 ± 32.0 ^a^
Rs	6.6 ± 0.12 ^b^	0.83 ± 0.030 ^b^	7.05 ± 0.52 ^b^	11.1 ± 0.20 ^b^	6.1 ± 0.10 ^a,b^	115.5 ± 9.5 ^b^
Rs + BC	9.5 ± 0.28 ^a^	0.94 ± 0.025 ^a^	10.20 ± 0.41 ^a^	16.2 ± 0.48 ^a^	6.7 ± 0.05 ^a^	380.0 ± 21.4 ^a^

Different letters indicate significant differences between means within columns at *p* < 0.05.

**Table 2 microorganisms-07-00676-t002:** Effects of biochar and *R. solanacearum* on soil available N, P, and K.

Treatment	Available N mg·kg^−1^	Available P mg·kg^−1^	Available K mg·kg^−1^
CK	420.0 ± 7.50 ^b^	128.8 ± 0.38 ^b^	162.8 ± 0.01 ^b^
BC	393.8 ± 6.25 ^b,c^	137.3 ± 0.56 ^a^	661.1 ± 0.08 ^a^
Rs	367.5 ± 5.05 ^c^	131.7 ± 0.85 ^b^	147.6 ± 0.04 ^b^
Rs + BC	520.0 ± 10.55 ^a^	139.8 ± 1.06 ^a^	736.8 ± 0.10 ^a^

Different letters indicate significant differences between means within columns at *p* < 0.05.

**Table 3 microorganisms-07-00676-t003:** Effects of biochar and *R. solanacearum* inoculation on soil microbial diversity.

Treatment	Shannon Index	Simpson Index	ACE	Chao
CK	6.51 ± 0.10 ^a^	0.99 ± 0.002 ^a^	6854 ± 517 ^a^	5590 ± 94 ^a^
BC	6.41 ± 0.12 ^a^	0.98 ± 0.002 ^a^	8805 ± 2561 ^a^	6674 ± 1401 ^a^
Rs	6.64 ± 0.06 ^a^	0.99 ± 0.001 ^a^	8370 ± 232 ^a^	6348 ± 179 ^a^
Rs + BC	6.61 ± 0.13 ^a^	0.99 ± 0.003 ^a^	9623 ± 1370 ^a^	6999 ± 794 ^a^

Different letters in the same column indicate significant differences among treatments (*p* < 0.05)

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
