# Peer review of "Biochar Suppresses Bacterial Wilt of Tomato by Improving Soil Chemical Properties and Shifting Soil Microbial Community"

_microorganisms, 2019, doi:10.3390/microorganisms7120676_

Round 1
Reviewer 1 Report
In this study, the authors estimated char made by wheat contributing to microbial community and soil chemical properties.
However, this reviewer could not fully pick up your main aims in this study. Although you already mentioned that there are a few studies for biochar effect on soil microbial community and mechanisms, enzyme activity, and pathogen, many studies have been reported similar research. Therefore, you have to highlight your unique results and aims.
Also, you selected and used wheat biochar. There are any scientific reasons? Cause, woods will be a good char on fertilization and plant cultivation.
This manuscript should have to edit by native speaker(s) for readability. Some parts were not easily understanding.
L19, replace by 16S rRNA gene
L65-66, w/w? also, you mentioned CN ratio was 48, but total C and N were 47 and 1%. Please check it.
L69, could you provide the define for soil organic matter? In the matters, organic C or N can be included.
L81-83, have to provide more information for used microorganisms such as strain name and 16S rRNA gene sequence.
L81-86, it seemed to be more importance in this study. For reproducibility, you should provide more detail. How many microbial cells were inoculated on each pot? Do you have a result for dosage dependent for biochar and/or microorganism? My recommendation is section 2.2 and 2.3 should be combined.
L104, 7d was 7days?
L103, part 2.5, 2.6 and 2.7 should be combined into one part.
L129, also, part 2.8, 2.9, and 2.10 should be combined into single part.
L136, replace by 16S rRNA gene. And also, please check and replace on whole manuscript.
L145, in line58-59, you used pyrosequencing. Which one was correct?
L151-165, you probably wanted to provide more information, but it’s too preliminary. More scientific contents seem to be necessary.
L171-176, sorry, this reviewer could not fully understand.
Fig1. Please define “dpi” and unit for y-axis.
L187-189, please don’t use %. Also, L187 double %%. Please rephrase your results based on scientific terminology.
Table 1, please define a, b, and ab
L193, this sentence seems to be more confusion.
L196-197, please see the comment for L187-189.
Table 2. again which means a, b, and ab?
L203-205, again… please don’t use % for fold… if you presented your activity, you should reanalyze and provide relative activities with figures.
Fig2. Again, define a, b, and ab
L215, how many reads and OTUs were used per each sample in this study? You should provide all information for read analysis and diversity indices by single table.
Fig3. It’s related to preliminary data. Delete.
L236, the result descriptions were preliminary. Need more polishing.
Fig4. How can we follow between your samples and diagram?
Fig5. How can you identify archaea using bacterial primer set? In addition, the abundances were also higher than other bacterial groups.
L253-256, sorry but I could not fully agree. The variation seemed to be nothing. Also, figures 6,7, and 8.
L290, sorry, your results were just showing that microbial diversity and community structure, and some enzyme activities, and soil chemical properties based on biochar treatment or not. If you discuss anything, you have use and interpret your results and the results’ relationship. However, in your discussion, it’s lack. E.g. in line 320-321, biochar could not affect to microbial diversity in this study. Why your biochar could not affect to them? Which is your idea? Also, in line 329-330, you mentioned corresponding result with other previous studies. It’s not interested. It has no differences.
L351, conclusions part should be combined into discussion
Author Response
Response to reviewer’s comments
Re:Manuscript Microorganisms-634176
Title:Biochar suppresses bacterial wilt of tomato through improving soil chemical properties and shifting soil microbial community
Authors:Yang Gao, Yang Lu, Weipeng Lin, Jihui Tian, Kunzheng Cai
Dear Editor and Reviewers,
We highly appreciate the valuable and critical comments of the editor and referees on our manuscript. The suggestions are quite helpful for us and we have made great modification in the new version of manuscript accordingly, and detailed corrections are listed below point by point. English language has been improved with the help of a native English speaker (Dr. Abdul Hafeez) expertise in soil ecology.
Revised portion is marked with color in the revised manuscript. We are looking forward to your consideration.
Thanks and Best Regards
Yours Sincerely
Kunzheng Cai

Reviewer 2 Report
This paper describes experiments documenting the effects of biochar amendments on bacterial wilt of tomato, soil physical and chemical properties, and soil bacterial communities, as well as some potential mechanisms responsible for disease suppression. It provides an appropriate and straight-forward design and presentation of data that strengthens and increases our understanding of the range and implications of the effects and impacts of biochar amendments and their role potential role in disease suppression and sustainable agriculture. Overall, the design and methodologies used are adequate and the analyses and interpretations are useful and supported. This paper provides useful and worthwhile information and will make a worthy contribution to the literature in this subject area. However, there are still some areas that need improvement before the paper is ready for publication.
Most importantly, the Materials & Methods and Results sections are not adequately detailed, leading to confusion or a lack of clarity or accuracy in some of methods and data presentation. Both sections are a bit too cryptic and more information and detail is needed. This must be improved prior to publication. Regarding methodologies, it is not clear how soil samples were collected (how much soil?, bulk soil, rhizosphere soil, etc.) and whether soil was collected from all pots in a replicate (5 pots?) and bulked into one composite sample, or whether soil only sampled from one pot per treatment replicate, or something in between? This may have important implications on the results, as if only sampled from one pot per replicate may not accurately represent what was going on across the different treatment pots. For some of the other analyses, methods not always clear. Have provided citations for methods, but in some cases, at least a brief description needed to clarify approach taken. Also, it is stated that experiments were repeated twice, but this can be ambiguous or misconstrued, because repeated twice actually means that the experiment was conducted 3 times, not 2, because if the experiment is repeated, it has been conducted twice, and thus repeated twice means it was conducted, then repeated, then repeated again. However, it is often used (incorrectly) to indicate that the experiment was repeated (conducted twice), thus it is important to clarify if the experiment was conducted two times or three times. If only twice, then should state that experiment was repeated (or conducted twice), but not repeated twice.
In the Results, First, it is not clear whether the results presented (in tables and figures) represent combined data from all experiments (2 or 3), or whether they are representative data from a single experiment. This is important to clearly state (If the same for all data, then can put into the M&M section, but if this varies, needs to be put in each table or figure as needed). Most importantly, some of the general descriptions in the Results text are not fully supported by the table and figure data, ansd more specific and info or clarifications are needed. For example, L273-275, states that the Biochar treatment significantly increased the abundance of the genus Flavisolibacter among others, however, although for most of those listed, the Rs+BC treatment did show a significant increase relative to Rs, this was not true for Flavisolibacter, which was only increased in BC, but not in Rs+BC. this should be clarified. In addition, several other Bacteroidetes genera were also increased in Rs+BC relative to RS, but were not mentioned, including Flavitalea, Adhaeribacter, and multiple unclassified Bacteroidetes. Need to be more careful to accurately and fully describe the relevant data in the text, and be specific about the observed results. Similarly, in lines 255-256, state that biochar did not influence the abundance of Protobacteria or Gemmatimonadetes, but the graphs clearly show that Rs+BC did increase abundance of both those groups relative to CK, just not relative to BC. Again, this should be clarified in text. Other places where there are problems with the data presentation are in lines 221-223, where it states that S10 had the highest OTU numbers, but according to the figure, this is not true (appears to be S12). Also, as for Shannon index numbers, these are not presented for individual samples, only for the treatment means, so those statements cannot be verified by the figures or tables. If wish to present data on the selected individual index values, need to include that data within the text (perhaps in Parentheses).
One other issue with the data presentation regards some problems with the figures themselves. Figure 2A,B: Not clear why y-axis is disrupted as the values appear to be continuous here, or should be continuous as there is not major break or gap to be depicted (use continuous axis). Figure 4: There is no legend for which treatments are represented by which symbols, thus it is impossible to determine which treatments are different from each other, negating the whole point of the figure. Please add legend. Figure 5: Legend depicts some 28 groups, yet only about 12 of those are actually discernable in the graph figure. This makes the legend very confusing and unhelpful. delete all the legend entries that are so small as to be invisible in the figure. Would be better to just include the top 12 or so groups that actually can be seen in the figure (remove others entirely). This would make the figure much more clear and useful (Archaea could be combined into one category, and other groups that are miniscule dropped from figure).
In addition to these presentation issues, there are several places where minor grammar or technical issues should be corrected or improved. These include such things as subject-verb agreement (L13-was/were), missing articles or modifiers (a, the, of), sentence fragments (L123-4), misspellings (L273-Bacteroidetes), spacing (L323), etc. The whole paper should be thproughly gone through for these and any other grammatical or technical errors or problems.
Once these revision/corrections are made the paper should be ready for publication.
Author Response

(The authors gave the same response as above.)

Round 2
Reviewer 1 Report
most concerns have been addressed. however, still, this reviewer could not understand why there are still denoted "*", "a", "b" or "ab" without any footnote in tables and figures.
Author Response
Response:
After significant analysis by data analysis software, a significant relationship between any two average values is obtained. All the average values are arranged from large to small, and the letter “a” is marked after the largest average. Use the largest average to compare with the second average, if it is significant, mark the letter “b” (ie. different letters); if the difference is not significant, mark the letter “a” (ie. the same letter).These letters reflect the 5% significant level, the 5 percent here represent the possibility of making a mistake by rejecting the hypothesis. The data marked with “a” and “b” (or c, d) are all different and significantly different from each other, but for comparison between “ef” and “fg” or “fg” and “gh”, there are repeated letters (“f” with “ef” and “fg”, “g” with “fg” and “gh”), so these two data are not significantly different. This analysis method is commonly and popular when comparing the difference among treatmentsand no necessary to denoted them with footnote.
And "*" indicates that there is a significant difference between the two sets of data at a significance level of 0.05 (p<0.05). For example,it can be marked as "*" under p<0.05,"**"under p<0.01, "***"under p<0.001.

Reviewer 2 Report
Authors have adequately addressed all of my previous comments, concerns, and suggestions. The manuscript should now be able to be accepted for publication.
However, there are still some minor errors/typos that should be cleaned up in the editing process. For example, L253 'most 11' should be transposed to '11 most' abundant; L284 plural agreement 'a further analyses' should be just 'further analyses' as analyses is plural; L308 misspelling 'soill'; and L344 insert 'an' before 'important'; and other minor edits, etc.